# Optimization strategy for sound environment of college basketball stadium

Jinyu Liang  *

Jinzhong University, Jinzhong, China

* liangjinyu888956@163.com

## Abstract

As an important sports venue, the sound environment of the basketball stadium is crucial for the experience of spectators and participants. In order to improve the quality of the acoustic environment of the basketball stadium, the acoustic parameters such as reverberation time, language transmission index and background noise were used to build a comprehensive evaluation system, and the software simulation was improved and optimized. The results show that the language transmission index of most measuring points in the basketball stadium has a high consistency between the on-site measurement values and the simulation results, and the simulation results are relatively accurate. The improved language transmission index values of each measurement point have significantly increased, with an average of 0.58, an increase of 93.33% compared to before the improvement. The improvement strategy effectively improves the acoustic environment of the basketball stadium and better controls the reverberation time in the stadium. The research provides a valuable analysis of the optimization of the sound environment of college basketball stadiums, which provides some reference for the improvement of the sound environment of basketball stadiums, and also has some reference significance for the optimization of the sound environment of other sports venues or entertainment venues. **Index terms** college basketball stadium; acoustic environment; optimization strategy; acoustic simulation; AHP.

## I. Introduction

As an important venue for hosting intense basketball games and mass sports activities, the acoustic environment inside the basketball stadium is crucial for the successful hosting of games and events. However, due to the parallel arrangement of the walls in the basketball stadium, the echo and reverberation phenomena generated have caused serious interference to the sound environment, posing difficulties for teaching and competition [1]. These issues may cause discomfort and distress to the audience and athletes, and even affect the progress and viewing experience

**Data availability statement:** All relevant data are within the paper.

**Funding:** The author(s) received no specific funding for this work.

**Competing interests:** The authors have declared that no competing interests exist.

of the competition. Therefore, in-depth research on the impact of the sound environment of basketball stadiums on user experience and proposing corresponding acoustic design strategies have positive significance for improving the sound environment quality of basketball stadiums and enhancing user experience [2]. The key method to solve problems such as long reverberation time (RT) and background noise interference is to optimize the sound environment, which can improve the sound propagation characteristics in the basketball stadium through scientific acoustic design and optimization. Among them, the application of Analytic Hierarchy Process (AHP) in the comprehensive evaluation of acoustic environment is crucial. Due to the involvement of multiple interrelated acoustic parameters in the acoustic environment, relying solely on a single parameter for evaluation is difficult to fully reflect the actual effect. AHP can decompose and compare these complex acoustic parameters in a hierarchical structure, quantify the impact of each parameter on the overall acoustic environment quality, and make the evaluation results more scientific, systematic, and accurate [3]. However, there are frequent issues of sound reverberation and noise in the acoustic environment of the basketball stadium, which affects the user experience and the multifunctional use of the venue. Therefore, the study uses AHP to construct a comprehensive evaluation system by incorporating acoustic parameters such as RT, language transmission index, and background noise, and improves and optimizes it through simulation using ODEON software. The research aims to improve the problems of excessive noise, improper acoustic reflection, and uneven sound diffusion in basketball stadiums. This study provides a systematic evaluation and optimization plan for the design and improvement of the sound environment in college basketball stadiums. This helps to enhance the user experience and functionality of university sports venues, providing theoretical support and practical reference for venue design and acoustic optimization. The study mainly includes four parts. Section I introduces the importance of the sound environment in basketball stadium and methods to improve its quality, and points out the specific methods of this study. Section II is a review of the sound environment and ODEON software. Section III is the ODEON acoustic environment simulation and optimization strategy of the college basketball stadium. Part A is the acoustic parameters of the acoustic environment of the college basketball stadium, and Part B is comprehensive evaluation system for sound environment of basketball stadium. Section IV is the analysis of acoustic environment optimization strategy of college basketball stadium. Among them, Part A is the analysis of measurement and analysis of acoustic parameters in college basketball stadium, and Part B is analysis of the effect of improving the sound environment in college basketball stadium. Section V is discuss divided into ODEON acoustic environment simulation and its optimization strategy of college basketball stadium.

## II. Related works

In recent years, with the emphasis on the stadium environment, research scholars have begun to focus on how to improve the sound environment of the stadium to provide a better experience and enhance the competitiveness of the competition. Shi W et al., for the performance evaluation of badminton stadiums, built a

computer program combining hierarchical analysis through MATLAB to determine the key performance indicators of the venues and calculate the importance weight of the evaluated performance attributes. The improvement strategy for badminton stadium facilities was ultimately determined [4]. Lee H M et al., in order to study the acoustic environment of the university campus, they used questionnaire survey and real-time noise level measurement to classify sound sources, evaluate the surrounding environment of the campus, and compared the noise levels of various sound sources. The university with the best acoustic environment quality was evaluated as a result [5]. Zhang D et al. established an evaluation system for building ventilation, sound, lighting, and other related indicators to assess the indoor environmental quality of sports facilities. The results indicate that ventilation indicators and noise treatment are key factors in ensuring indoor quality [6]. Jung C et al., in order to evaluate the indoor environment of residential buildings, researchers completed the comprehensive evaluation according to the post-occupancy assessment survey and the weight of indoor environmental factors, and determined the priority of residents' space needs. It was found that thermal comfort factors in summer and indoor air quality factors in winter are more important [7]. Di Loreto S et al. developed a tool for predicting classroom speech transmission metrics in order to evaluate the comprehensibility of speech in the classroom. It was found that this method can be applied to indoor sound environment design and has certain computational robustness [8]. Zhang L et al. designed a system using acoustic equipment and video sensors to evaluate indoor gymnastics activities and improve activity quality. The results indicate that the system can alleviate the data imbalance between acoustic and perceptual environments [9].

As an acoustic simulation tool, ODEON software has been widely used to improve the acoustic environment in various venues and show its potential in the gymnasium field. Murgia S et al. used ODEON software to analyze the indoor environment of elementary school classrooms and conducted comprehensibility tests. It was found that long RT and high background noise can have a negative impact on speech transmission [10]. Esmaeil and Salehi, in order to study the variables affecting the receiving sound intensity, professionals used the full factor method to identify the target scene of the variables, and used ODEON software to determine the frequency range with the greatest influence on the sound intensity. It was found that the frequency of sound and the distance between the sound source were inversely proportional to the sound intensity, and the octave band of sound had the greatest impact on the sound intensity [11]. Labia L et al., in order to improve the acoustic quality of the conference room, the team used ODEON software to complete the acoustic measurement of the conference room, simulate sound absorption and diffusion processing. Afterwards, determine the optimal number and location of acoustic panels indoors, and use absorbent materials appropriately to reduce costs [12]. Lv C et al. developed a spatial acoustic simulation system that combines Unity 3D and ODEON software to achieve model visualization. The results show that the system has good visualization effect and operability, and the absolute error of acoustic parameter values is small [13]. Sanz Soriano et al., for stage acoustics and parametric design, they enhanced the design collaboration in the early design stages through ODEON software and 3D modeling platform, and evaluated the impact of architectural changes on the reflection of supporting the orchestral ensemble. The different division of labor in these software can achieve real-time feedback and better integrate architectural and acoustic design [14]. Iannace G et al., in order to carry out the indoor sound school, they introduced the sound school system made of sound-absorbing flexible board to measure the acoustic characteristics of the empty room, and used ODEON software to analyze the acoustic characteristics and analyze the room configuration. This test found that installing 1.5 mm soft wooden boards 3.0 cm away from the room wall is reasonable [15].

In conclusion, ODEON software has shown great potential in improving the sound environment of sports venues. However, using acoustic software alone is not enough to comprehensively analyze the acoustic environment of basketball stadiums. Therefore, the study adopts the AHP to integrate acoustic parameters such as RT, language transfer index, and background noise, and establish a comprehensive evaluation system. And use ODEON software for simulation to improve and optimize the acoustic environment. The research aims to reduce interference such as noise and echo, improve the clarity and quality of sound, and control noise levels. Thus providing actionable technical solutions for optimizing the

acoustic environment of university basketball stadiums and promoting the improvement of acoustic quality in the construction of university sports facilities.

## III. Sound environment simulation and its optimization strategy of college basketball stadium

To measure the acoustic environment of the basketball stadium, build a comprehensive evaluation system for the obtained acoustic parameters, and improve and optimize it with ODEON software.

### A. Acoustic environment parameters of college basketball arena

To study the sound environment of the college basketball stadium, the starting point and the end point of the selected sound frequency range are 125 Hz and 4000 Hz, respectively. Because the 125 Hz low frequency sound is usually associated with low sound reflection and resonance in the basketball arena. The special structure and large space of the basketball arena may lead to the accumulation and reflection of low-frequency sound, having an impact on the propagation of sound and the distribution of the sound field. At the same time, in the basketball arena environment, the 4000 Hz high-frequency sound may be attenuated by various factors, such as air absorption, surface scattering and absorption. Moreover, humans have relatively low sensitivity to high frequency sounds, so the main focus is on the low frequency and intermediate frequency range [16] when studying the basketball arena acoustic environment. In order to facilitate the experiment, the audible frequency range is divided into the frequency band, and each band contains the upper and lower frequency. If the upper frequency $f_s$ and the lower frequency $f_x$ of one band $n$ are different, as shown in Equation (1).

$$2^n = \frac{f_s}{f_x}$$

(1)

In general acoustic studies, the frequency band and 1/3 frequency band are divided according to the central frequency $f_z$ size order, as shown in Equation (2).

$$f_z = \sqrt{f_x \cdot f_x}$$

(2)

The frequency band is used in the acoustic environment of the college basketball stadium, $f_z$, $f_s$ and $f_x$ is as shown in Table 1.

The room pulse response (Room Impulse Response, RIR) refers to the measurement curve of the resulting sound signal over time after emitting a short sound pulse in a closed space. It contains the acoustic properties of the room, such as RT, clarity, and so on. Obtaining the room pulse response of the basketball arena can provide important reference data for acoustic design and adjustment to ensure that the acoustic environment in the gym meets the requirements and provide a good auditory experience. According to linear system theory, if an acoustic system in a closed space is linear, the output signal $R(t)$ at the receiving point can be calculated from the sound source signal $S(t)$ [17]. $R(t)$ and $S(t)$ relationship expression, as shown in Equation (3).

$$R(t) = S(t) * G(t) = \int_{-\infty}^{+\infty} S(\tau) G(t-\tau) d\tau$$

(3)

**Table 1. Center frequency and upper and lower limit frequencies of the octave band.**

| $f_z$ (Hz) | 125 | 250 | 500 | 1000 | 2000 | 4000 |
|---|---|---|---|---|---|---|
| $f_s$ (Hz) | 178 | 355 | 709 | 1414 | 2822 | 5630 |
| $f_x$ (Hz) | 89 | 178 | 354 | 707 | 1411 | 2815 |

In equation (3), $G(t)$ is RIR. RT is an important indicator to measure the attenuation speed of reflected sound in the indoor acoustic environment. It is defined as the time required for the sound to continuously reflect, attenuate and disappear in the indoor environment after the sound source is turned off. To measure RT, the T60 indicator is usually used, the time required for the sound to decay to 1 in 60 of the original sound intensity. A larger T60 value indicates the longer RT and vice versa. Basketball stadiums generally have a large volume $V$, and at this time, the air has a greater absorption capacity for sounds with frequencies above 2000Hz. RT expression, as shown in Equation (4).

$$RT = 0.161 \frac{V}{-S \ln(1 - \bar{\alpha}) + 4mV}$$

(4)

In formula (4), $S$ is the total of the internal surface area of the room, $\bar{\alpha}$ is the average sound absorption coefficient in the room, and $4m$ is the air attenuation coefficient. Where $\bar{\alpha}$ expressed, as shown in Equation (5).

$$\bar{\alpha} = \frac{\alpha_1 S_1 + \alpha_2 S_2 + \ldots + \alpha_n S_n}{S_1 + S_2 + \ldots + S_n}$$

(5)

Where $m$ expressed, as shown in Equation (6).

$$m = \frac{170 f^2}{\psi} \times 10^{-4}$$

(6)

In Equation (6), $\psi$ is the air humidity, $f$ is the sound frequency. Language clarity $D_{50}$ and language transmission index (speech transmission index, STI) are commonly used objective evaluation indicators. Language clarity is the proportion of energy in the key frequency range associated with speech comprehension relative to total energy when the sound travels to the listener's ear in an indoor environment. Language clarity $D_{50}$ of expression, as shown in Equation (7).

$$D_{50} = \frac{\int_0^{50ms} p^2(t)\, dt}{\int_0^\infty p^2(t)\, dt}$$

(7)

In Equation (7), $p(t)$ is the pulse response of the receiving point. STI is an index to measure the quality of voice transmission. It is based on the spectral characteristics of voice signals and the influence of background noise, and comprehensively considers the distortion, noise, reverberation and other factors of voice signals in the transmission process. The STI values range from 0 to 1, and the closer the value is to 1, the better the speech transmission quality and the higher the speech clarity. Modulation transfer function (Modulation transfer function, MTF) is a function used to describe the response characteristics of the transmission system to different modulation frequencies. It has some association with STI, because the modulation characteristics of speech signal will be considered in the calculation process of STI. The MTF expression, as shown in Equation (8).

$$m(f_m) = \left| \int_0^\infty h^2(t)\, e^{-j2\pi f_m t}\, dt \right| \Big/ \left( \int_0^\infty h^2(t)\, dt \right) \cdot \left( 1 + 10^{-SNR/10} \right)^{-1}$$

(8)

In Equation (8), $f_m$ is the modulation frequency, $h(t)$ is the room pulse response and $SNR$ is the room signal to noise ratio. The ISO International Organization for Standardization (International Organization for Standardization, ISO) defines the noise evaluation curve (Noise Rating, NR) curve for the indoor background noise evaluation. The NR curve is used to decay the low-frequency noise with A frequency weight to simulate the perceptual characteristics of the human ear. The NR curves give a series of standardized rating values at different frequencies, usually in dB. These values can be used

to compare the indoor background noise levels of different rooms, or to assess whether the noise in a room meets the specific requirements [18].

The allowable A sound level noise limit of the basketball arena is 45dB. Then, the CEL-24X sound level meter is used to measure the background noise. The measurement method is to fix the sound level meter at the vertical height of 1.5m from the ground, and then placed in the center of the basketbal stadium after the layout. At this time, the sound level meter switch is in dB (A) and Slow gear. Data were recorded once at 5s, repeated to 200, and then data were sorted in a large to small manner. Finally, the equivalent sound pressure level calculation $L_{eq}$ is completed, as shown in Equation (9).

$$L_{eq} = L_{50} + (L_{10} - L_{90})^2 \big/ 60$$

(9)

In equation (9), $L_{10}$, $L_{50}$ and $L_{90}$ are the 20th, 100th and 180th data, respectively, indicating the A sound level over 10%, 50% and 90% of the measurement time, respectively.

## B. Comprehensive evaluation system for sound environment of basketball stadium

In the simulation analysis of the acoustic environment of college basketball stadium, various acoustic parameters need to be determined and considered first. These acoustic parameters can help to evaluate the acoustic performance in the basketball arena, and thus optimize the acoustic environment design of the basketball arena. In order to eliminate the trembling echo of the basketball stadium, the RT is shortened to improve the language clarity, and the sound absorption is processed to optimize the sound environment in the stadium. In order to improve the calculation efficiency and accuracy, ODEON is used for simulation. ODEON simulation simulation design flow diagram, as shown in Fig 1.

In Fig 1, in order to facilitate operation, the basketball stadium model, and then use ODEON programming to build the internal model. Later, the surface of the model is distorted and overlapped. If so, it is necessary to modify the model and reprogram it. If not, start setting the sound source and reception point. After the layout is completed, add materials to the internal surface consistent with the internal measurement. To determine whether the sound waves are limited by the closed environment, and whether there is any leakage or reflection, the closed case of the model is detected by acoustic particles and acoustic lines. Acoustic particle detection is a particle-tracking-based method that simulates the propagation of sound waves in the model. By releasing a set of acoustic particles in the simulation and tracking their propagation path, it is possible to determine whether the sound waves can be completely enclosed by the model. If the acoustic particle encounters leakage or reflection during the simulation, this may mean that the model has a problem with incomplete closure. Acoustic line detection is a method based on acoustic line propagation, a virtual line along the propagation path of the acoustic wave, which can detect the closure of the acoustic wave in the model. If the acoustic line encounters any

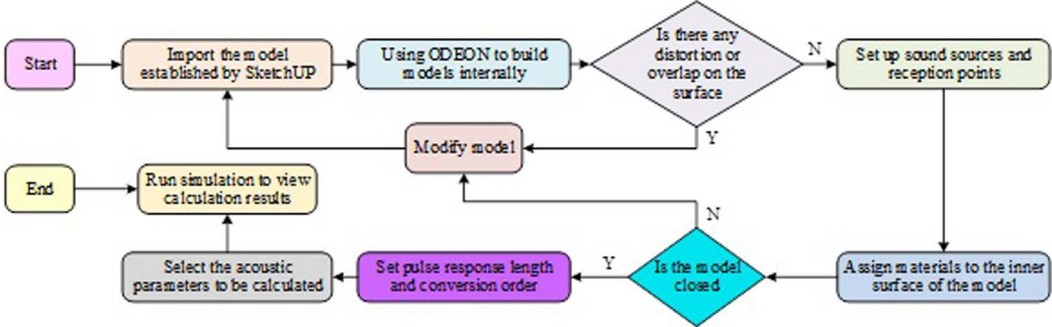

**Fig 1. Schematic diagram of ODEON simulation design process.**

leakage or reflection in the model, then the closure of the model may be problematic. After detecting the model closure situation, considering the accuracy of the sound field simulation and the effect of the results, the setting of the room parameters is very important [19]. The study sets the pulse response length and the conversion order, selects the acoustic parameters to be calculated, runs the simulation, and views the calculation results. The study used the 3D modeling software Sketchup to build a basketball stadium model, ensuring that the size, layout, and shape of the internal space accurately reflect the actual venue. The distribution of measuring points (measure point, MP) in the basketball stadium is shown as shown in Fig 2.

In Fig 2, the size of the basketball arena selected for the study is 40m and 54m, and the volume of the field area is about 76,000 m³. There are 32 measurement points set up throughout the stadium. In order to ensure that sound waves can evenly cover the entire venue and simulate a standard sound field distribution. Therefore, this study chose the sound source to be located at the center. This can simplify the design of experiments and data analysis. By keeping the sound source position unchanged and paying attention to the effects of other variables such as materials, sound absorption characteristics, and reflections, the complexity caused by the sound source position can be avoided. This helps to analyze the direct impact of different materials or improvement measures on acoustic effects, without being affected by other changing factors. Study selection hierarchy analysis, each indicator is determined by the weight occupied in that evaluation. A Schematic diagram of the comprehensive evaluation hierarchy of the basketball stadium, as shown in Fig 3.

In Fig 3, the comprehensive evaluation hierarchy model contains three levels. RT, STI and $L_{eq}$ evaluation indicators are used as the criterion layer to build a comprehensive evaluation system for the acoustic environment of the basketball stadium, so as to obtain the acoustic environment evaluation level. Then a pairwise comparison is made to determine the matrix A construction. The matrix form is shown in Equation (10).

$$A = (a_{ij})_{n \times n} \tag{10}$$

In equation (10), $a_{ij}$ is the element ratio $A_i$, $A_j$ is the proportion of the importance of an element in the previous layer, that is, the importance amplitude is measured according to the scale of 1–9, and its properties are, and,. Study the consistency of the matrix, first with the consistency index as shown in Equation (11). $a_{ij} \geq 0$ $a_{ij} = 1/a_{ji}$ $i \neq j$ $a_{ii} = 1$ CI

$$CI = (\lambda_{max} - n)/(n - 1) \tag{11}$$

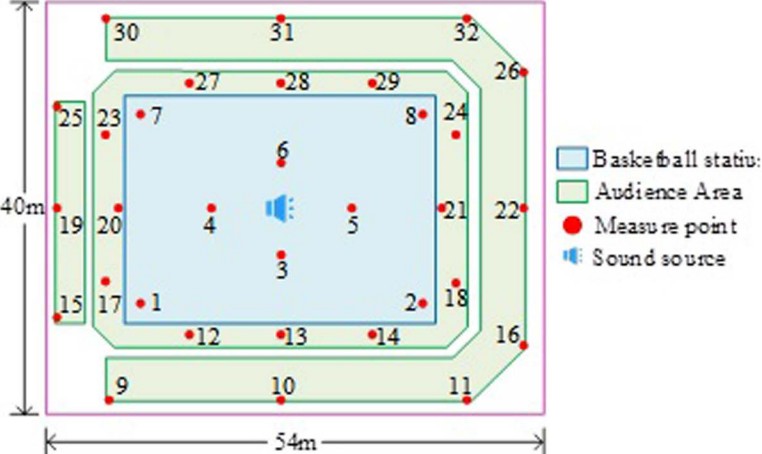

**Fig 2. Distribution diagram of measuring points.**

In Equation (11), $\lambda_{max}$ the largest characteristic root of A, $n$ is the order A [20]. Conformity index RI, consistency ratio CR, as shown in Equation (12).

$$CR = {CI}/{RI}$$

(12)

Considering the relative importance of the elements, the judgment matrix A of the criterion layer elements is shown in Table 2.

The weight coefficient of each layer index to the target layer is the set average of the elements in each line of the matrix, as shown in Equation (13).

$$\bar{W}_i = \sqrt[n]{\prod_{j=1}^{n} a_{ij}} \, (i,j = 1, 2, ..., n)$$

(13)

In Equation (13), $\overline{W} = \left[\overline{W}_1, \overline{W}_2, ..., \overline{W}_n\right]^T$ is the normalized treatment vector, $W_i$ is the weight of each indicator, as shown in Equation (14).

$$W_i = \frac{\overline{W}_i}{\sum_{j=1}^{n} \overline{W}_j}$$

(14)

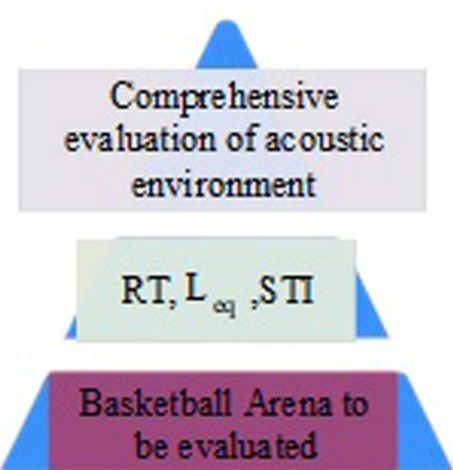

**Fig 3. Schematic diagram of the hierarchical structure of the comprehensive evaluation of the basketball stadium.**

**Table 2. Judgment matrix of criteria layer elements.**

| Target layer | RT $A_1$ | STI $A_2$ | $L_{eq}$ $A_3$ |
|---|---|---|---|
| RT $A_1$ | 1 | 3 | 5 |
| STI $A_2$ | 1/3 | 1 | 3 |
| $L_{eq}$ $A_3$ | 1/5 | 1/3 | 1 |

In Equation (14), $W_1 = 0.6370$, $W_2 = 0.2583$, $W_3 = 0.1047$. After the weight allocation table of each indicator, the RT, STI and $L_{eq}$ evaluation indicators of the criterion layer are scored respectively, and the comprehensive score M of the sound environment evaluation of the basketball stadium, as shown in Equation (15).

$$M = 0.63704A_1 + 0.2583A_2 + 0.1047A_3 \qquad (15)$$

## IV. Analysis results of the sound environment optimization strategy of college basketball arena

This study first conducted preliminary measurements of the existing acoustic environment of university basketball stadiums, obtaining acoustic parameters such as RT and STI. Next, the effects of different sound-absorbing materials in the stadium were analyzed to predict and verify the improvement effects of these materials. Then, based on the selected materials, sound particle and sound line detection were carried out inside the stadium to simulate sound wave propagation and evaluate the actual effect of the materials. Finally, by integrating all the data, the comprehensive score of the sound environment evaluation of the basketball stadium was calculated, and the final acoustic parameters of the stadium were obtained.

### A. Measurement and analysis of acoustic parameters in college basketball stadium

CATT-Acoustic is an advanced acoustic simulation software used to predict and optimize the performance of sound systems. It has the characteristics of high precision and flexibility, but the operation is complex and requires high professional knowledge. Enhanced Acoustic Simulator for Engineers (EASE) has lower accuracy than ODEON and lacks the function of a comprehensive evaluation system. The proposed strategy has significant advantages in comprehensive evaluation and precise simulation, and is suitable for complex acoustic environment improvement projects that require high-precision and comprehensive analysis. But its complexity and cost are high, making it suitable for projects with sufficient budget and technical support. This study selected basketball stadiums that met the research criteria and carried out acoustic renovations based on the improvement measures in the simulation. Use professional acoustic equipment to measure before and after renovation, collect data such as RT, language transfer index, and background noise. Compare and analyze the actual measurement results with the ODEON simulation results to verify the accuracy of the simulation and the effectiveness of improvement measures. Comparison of the results of the field measurement RT of different MP's in the basketball arena, and the clarity of the field measurement language in the basketball arena $D_{50}$ The mean values, and the mean results of the RT field measurements and simulation values, are shown in Fig 4.

In Fig 4(a), the RT results of MP at 125 Hz are less consistent and more scattered. This is because low-frequency sound waves are easily affected by various factors such as reflection, standing waves, and sound wave diffraction when propagating in large spaces. The selected MP1, MP2, MP7, and MP8 are located in the corners of the rectangular field, and the positions of the other MPs also exacerbate this unevenness, resulting in significant differences in RT measured at different locations. Above 250 Hz, the RT measurement results of different MP's are similar, indicating that the RT in the basketbal stadium has a good uniformity, and the results have certain reliability. In Fig 4(b), the field measurement $D_{50}$ at the frequency is 250Hz The mean value was 0.13, the mean of field measured RT was 4.12s and the mean of simulation RT was 4.56s. $D_{50}$ at 4000Hz The mean was 0.28, the mean simulation RT was 2.48s, and the mean field measured RT was 3.18s. At this time, there are obvious differences between the field measurement and the simulation results, with a difference of 0.70s on-site measurement $D_{50}$. The average value of the decay in each frequency band is very fast, which may lead to unclear language and low accessibility. During the simulation process, the sound absorption coefficient used does not completely match the performance of the sound absorption material in the actual scene, which may lead to errors between the simulation results and the field measurement results.

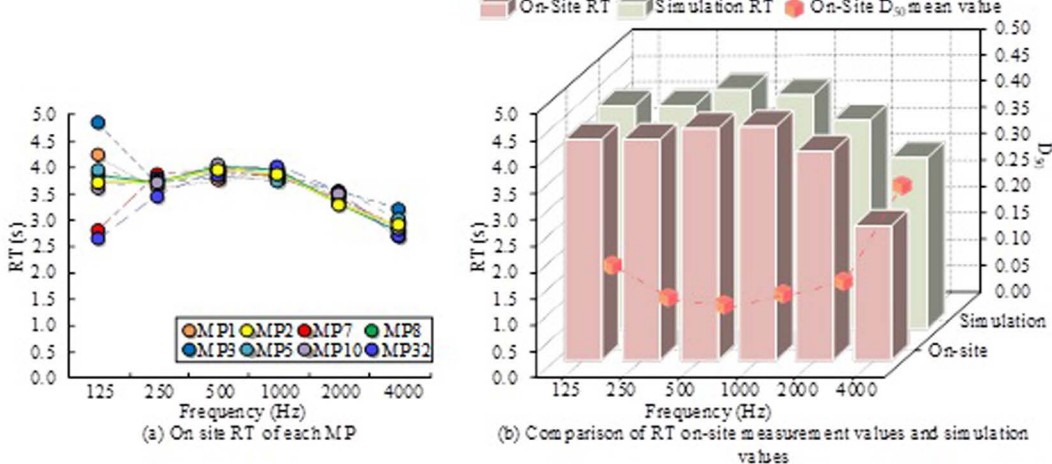

**Fig 4. Comparison of RT and D50results.**

In order to facilitate the pulse response measurement of the basketball stadium, a large number of balloons were prepared. The pulse sound source was the sound of the broken balloon. In the case of the empty field, the data was collected using the voice recorder and other measuring equipment, and then the collected acoustic signals were imported into DIRAC software to analyze the acoustic parameters. Among them, the room pulse response of the basketball arena is shown in Fig 5.

In Fig 5, since the competition hall of the basketball stadium should meet the requirements of NR-40 limit, the noise limit is 45dB, and the equivalent sound pressure level of the background noise of the basketball stadium is 40.74dB, so the basketball stadium meets the standard limit requirements.

The setting accuracy of ODEON simulation remains unchanged. The pulse response length of the basketball stadium is 4500ms, the conversion order is set to 2, the number of sound lines is 10000, and the scattering coefficient is 0.05. The perforated steel plate and wood fiber board were selected to improve the wall of the basketbal stadium. The cavity depth between the perforated steel plate and the wall is 150 mm, with 2% perforation rate, 5 mm perforation diameter,

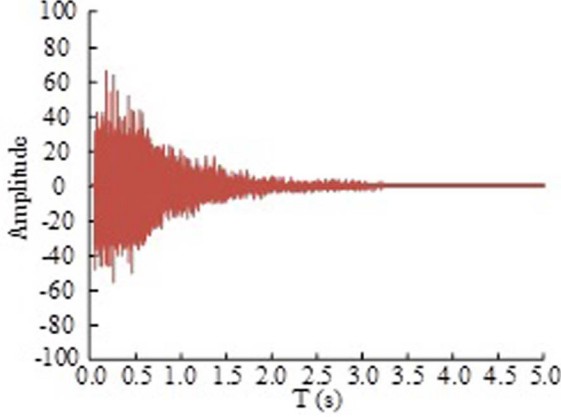

**Fig 5. The room pulse response of the basketball stadium.**

plate thickness is 1 mm, and the filling weight of medium frequency sound is 25 kg/m³Superfine glass wool. The perforated gypsum board ceiling is installed on the top surface of the venue, with 200 mm cavity behind the board, with the perforation rate of 6% and 7 mm plate thick. The measured mean value and estimated value of the RT empty field, as well as the sound absorption coefficient of different materials, are shown in Fig 6.

In Fig 6(a), RT data at different frequencies were generated through ODEON simulation. At a frequency of 500 Hz, the average values of full field estimation and empty field measurement are 3.84s and 4.45s, respectively. The difference between these two values represents the variation in the sound absorption performance of the material. At 4000 Hz, due to the strong sound absorption effect of wood fiberboard and perforated steel plate, the RT value is relatively low. At this time, the estimated value of the entire field and the average value of the empty field measurement are 2.84s and 3.18s, respectively. In Fig 6(b), the sound absorption coefficient of wood fiberboard in the range of 125 Hz~250 Hz is between 0.2~0.3. Its sound absorption coefficient in the range of 500 Hz~1000 Hz is between 0.4~0.7. Its absorption coefficient is highest in the range of 2000Hz~4000 Hz, between 0.7~0.9. The sound absorption coefficient of wood fiberboard at different frequencies directly affects the RT of the basketball stadium. By absorbing mid to high frequency sound waves, the wood fiberboard significantly reduces RT, keeping the RT value within a reasonable range between 500 Hz-4000 Hz. The problem of low low-frequency sound absorption coefficient is compensated for by combining perforated steel plates and ultrafine glass wool, achieving full frequency RT optimization. The perforated steel plate has strong impact resistance, high strength and stiffness. The wall sound absorption material can effectively reduce noise reflection in basketball stadiums and improve the quality of the acoustic environment. At the same time, these sound-absorbing materials can also provide some protection for walls and columns, avoiding wear and tear due to long-term use. Perforated steel plates have strong durability and low maintenance costs. Its impact resistance can protect walls and columns, extending their service life. Economically, high initial investment is required for materials and construction costs, but the long-term maintenance cost of this method is relatively low, and it can provide significant acoustic environment improvement for basketball stadiums.

To determine whether the sound waves are limited by the closed environment, and whether there is any leakage or reflection, the closed case of the model is detected by acoustic particles and acoustic lines. Simulate the propagation path of sound rays in the model in ODEON software, and release a set of acoustic particles in the simulation and track

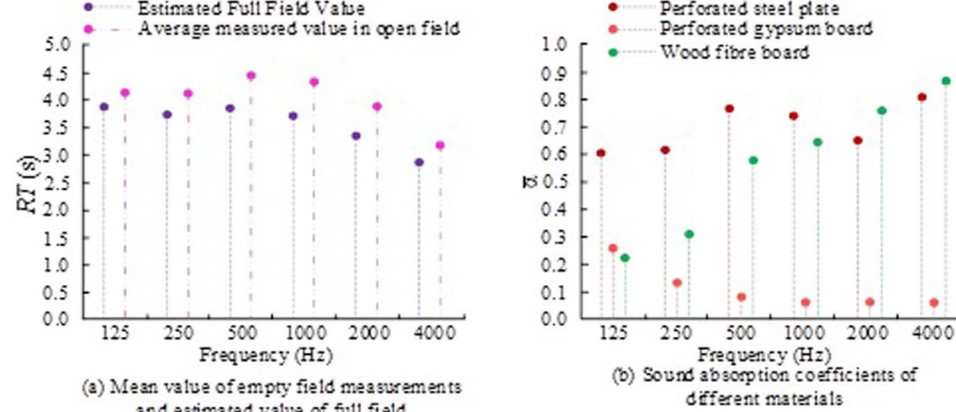

**Fig 6. Comparison of RT and sound absorption coefficient results at different frequencies.**

their propagation path. The results of acoustic line detection and acoustic particle detection in basketball stadium are shown in Fig 7.

In Fig 7(a), the acoustic lines inside the model do not escape or leak into the environment outside the model. This closure is very important for the acoustic simulation and simulation, which can ensure the accuracy and reliability of the simulation results. In Fig 7(b), the propagation and decay of the acoustic particles inside the model do not have a detectable effect on the external environment. The fully closed model can provide stable and reproducible results.

## B. Analysis of the effect of improving the sound environment in college basketball stadium

Install sound level meters at each measurement point to ensure they are not affected by external environmental noise and maintain a distance of 1.5m from the ground. Within 30 minutes, use a sound level meter to continuously measure the sound pressure level inside the basketball stadium to capture changes in environmental noise. Based on the measurement results, calculate the $L_{eq}$ at each position, which is the average sound pressure level during the measurement period. The $L_{eq}$ results of each measurement point are averaged to obtain the overall equivalent sound pressure level within the venue. The RT, STI and $L_{eq}$ index scoring standards before and after the improvement are shown in Table 3.

In Table 3, when the basketball stadium is empty, the score is 1 point for $L_{eq}$ >50.00, 2 points for 45.00dB< $L_{eq}$ <50.00dB, and 3 points for $L_{eq}$ ≤ 45.00dB. Before the improvement, the scores for $A_1, A_2$, and $A_3$ were 1, 1, and 3, respectively. After the improvement, the scores for $A_1, A_2$, and $A_3$ were 3, 2, and 3, respectively. When 1.00 ≤ $M$ =1.21 ≤ 1.50, the acoustic environment evaluation of the basketball stadium before improvement is poor. When 2.50 ≤ $M$ =2.74 ≤ 3.00, the sound environment evaluation of the improved basketball stadium is good.

In order to evaluate the effect of the improvement of the basketball arena and whether it meets the requirements of the international standards, the study compares the RT value before and after the improvement with the international limit when the basketball arena is full, as shown in Fig 8.

In Fig 8, when the frequency is 125 Hz and 250 Hz respectively, the international limit of the basketball arena is 3.25s and 3.00s respectively, and the frequency band is 500 Hz, 1000 Hz, 2000Hz and 4000 Hz, the international limit is 2.50s.

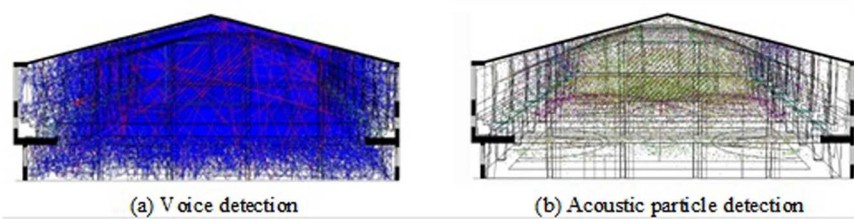

(a) Voice detection    (b) Acoustic particle detection

**Fig 7. Comparison of sound line detection and particle detection results in basketball stadium.**

**Table 3. Results of indicator scoring standards.**

| Status | Before improvement | | | Improved | | |
|---|---|---|---|---|---|---|
| Index | RT (s) $A_1$ | STI $A_2$ | $L_{eq}$ (dB) $A_3$ | RT (s) $A_1$ | STI $A_2$ | $L_{eq}$ (dB) $A_3$ |
| Score | 3.76 | 0.30 | 40.74 | 1.64 | 0.58 | 40.74 |
| $A_1$ score | 1 | / | / | 3 | / | / |
| $A_2$ score | / | 1 | / | / | 2 | / |
| $A_3$ score | / | / | 3 | / | / | 3 |
| $M$ | 1.21 | | | 2.74 | | |

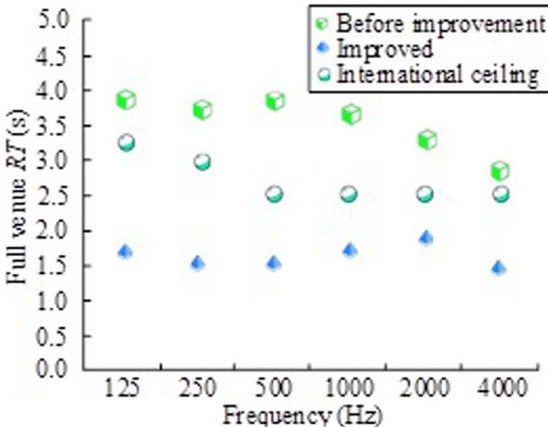

**Fig 8. Comparison of results before and after improvement with the international upper limit RT value.**

Before the improvement, the full field RT is greater than the international limit in each frequency band, which exceeds the international limit of 500 Hz and 1000 Hz by 1.30s and 1.10s respectively. However, the improved full RT is significantly less than the international limit, and the lowest full RT at 250 Hz is 1.59s, and the highest full RT at 2000Hz is 1.95s. Shorter RT reduces reverberation and reverberation of sounds, allowing listeners to hear sounds more clearly, improving sound quality and intelligibility. At the same time, the improvement strategy effectively improves the acoustic environment of the basketball library, and better controls the RT in the library. The results before the improvement are the results of the field measurement, comparing the STI before and after the improvement of the basketball stadium and the simulated STI results, as shown in Fig 9.

In Fig 9, there is a small error between the STI field measurement value and the simulation value of the 32 measurement points set in the basketball arena, with a small result value. The STI field measurement values and simulation value results of most measurement points are distributed around 0.3, and the two curves have a high consistency, which indicates that there are more accurate simulation results. The improved STI value of each measurement point increases significantly, and the average value is 0.58, which is 0.28 compared with the improved STI average of 0.3, indicating that the speech signal is clearer and more accurate in the transmission process.

## V.  Discussion

This study used the AHP combined with RT, STI, and other acoustic parameters to establish a comprehensive evaluation system, which effectively improved the acoustic environment of the basketball stadium. The results indicate that the improved RT values are below the international standard upper limit in all frequency bands, especially at 250 Hz where the improvement effect is most significant, with RT reduced to 1.59s. This indicates that the RT of the basketball stadium has been significantly controlled, reducing reverberation and making the sound clearer and more distinguishable. The efficient absorption of mid to high-frequency sound waves by wood fiberboard significantly improved the STI, from 0.30 to 0.58. This further proves that the clarity of speech in the basketball stadium has been significantly improved. Yang D and other professionals used non-linear regression models to evaluate STI and RT values in classroom interiors, obtaining relatively accurate data [21]. In contrast, this study not only measured the acoustic environment of the basketball stadium, but also verified the results of STI through simulation, and achieved comprehensive improvement of the acoustic environment through optimization of multi band RT. The comprehensive score of 2.74 indicates the inadequacy of the basketball stadium's sound environment before the improvement, while the evaluation result after the improvement is favorable,

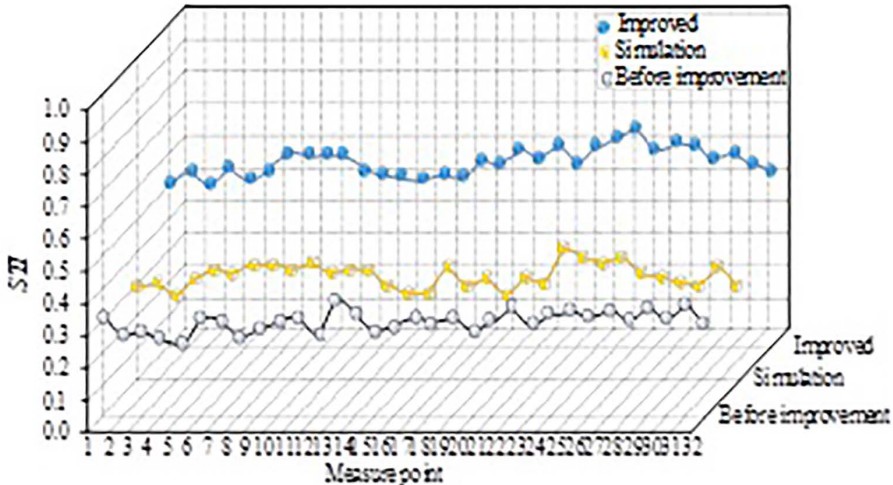

**Fig 9. Comparison of STI results before and after basketball stadium improvement and simulated STI results.**

showing that the measures taken have effectively enhanced the acoustic quality. The AHP used in this study not only considers RT and STI, but also integrates acoustic parameters to establish a more comprehensive evaluation system, resulting in a more systematic optimization of the acoustic environment in the basketball stadium. In large spaces such as basketball stadiums, measuring sound pressure levels helps evaluate the intensity of sound. Before improvement, the scores for $A_1, A_2$, and $A_3$ were 1, 1, and 3, respectively. At this time, the comprehensive score is M =1.21, which is within the range of $1.00 \leq M \leq 1.50$, and the sound environment evaluation of the basketball stadium is poor. After improvement, the scores of $A_1, A_2$, and $A_3$ have been increased to 3, 2, and 3, respectively. At this time, the comprehensive score is M =2.74, which is within the range of $2.50 \leq M \leq 3.00$. The sound environment evaluation of the basketball stadium is good. Through improvement measures, the sound environment quality of the basketball stadium has been significantly improved.

Compared with other studies, Zhang C et al. proposed that fiber materials can achieve good sound absorption performance for reducing noise pollution [22]. The wood fiber board selected as the sound-absorbing material in this study not only has good sound absorption performance, but also plays an important role in the acoustic improvement of the basketball stadium, effectively reducing noise and reverberation, and improving the quality of the sound environment. In addition, the research method of Siregar I team is to use software and trial and error analysis to evaluate university auditoriums, and to use different acoustic materials for sound insulation design. The results indicate that this method can comprehensively adjust the indoor acoustic environment and ensure the functionality of indoor acoustics [23]. Similarly, this study conducted scientific simulations using ODEON software to carry out professional construction, ensuring the effectiveness and reliability of the sound environment renovation in the basketball stadium. From the perspective of improving the quality of the acoustic environment and the durability of facilities, the wood fiberboard and Perforated steel plates used in this study have high value and feasibility. Baquero et al. proposed multiple evaluation indicators for urban acoustic environment perception, including soundscape, noise annoyance, sensitivity, and acoustic comfort, which are of great significance for the overall evaluation of the acoustic environment [24]. The comprehensive evaluation system of this study also considers multidimensional acoustic parameters. By optimizing RT and STI, the sound environment in the basketball stadium has been successfully improved, achieving higher comfort and clarity standards, and verifying the practicality and effectiveness of this method in practical applications.

## VI. Conclusion

In order to improve the clarity and intelligibility of voice transmission in the basketball stadium, RT, STI and $L_{eq}$ acoustic parameters were used to build a comprehensive evaluation system, and the ODEON software was used to improve and optimize to reduce noise and reverberation. The results show that the estimated mean value of RT full field and the smallest at 4000 Hz are 2.84s and 3.18s respectively. RT was highest at 500 Hz at 3.84s and 4.45s, respectively. Perforated steel plate has strong impact resistance, high strength and stiffness, as a wall sound absorption material can protect the wall and column from the impact of basketball, and provide better sound absorption effect. Fiberboard has good sound absorption performance, can absorb and convert sound waves into weak heat energy, used for wall decoration, and reduce the generation of echo. Since the competition hall of the basketball stadium should meet the requirements of NR-40 limit, the A sound level noise limit is 45dB, and the equivalent sound pressure level of the background noise of the basketball stadium is 40.74dB, the basketball stadium should meet the standard limit requirements. The acoustic lines inside the model do not escape or leak into the environment outside the model. This closure is very important for the acoustic simulation and simulation, which can ensure the accuracy and reliability of the simulation results. The propagation and decay of sound particles inside the model will have no detectable effect on the external environment. The fully closed model can provide stable and reproducible results. But the study did not consider the weakening effect of sound-absorbing materials in basketball stadiums after long-term use. In the future, intelligent monitoring systems can be designed to monitor the sound environment inside the stadium in real time, in order to intervene or solve problems.

## Author contributions

**Conceptualization:** Jinyu Liang.

**Data curation:** Jinyu Liang.

**Formal analysis:** Jinyu Liang.

**Investigation:** Jinyu Liang.

**Methodology:** Jinyu Liang.

**Project administration:** Jinyu Liang.

**Software:** Jinyu Liang.

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
