## [Decision Letter · Decision Letter 0]

11 Feb 2025

Dear Dr. Liang,

Thank you for submitting your manuscript to PLOS ONE. After careful consideration, we feel that it has merit but does not fully meet PLOS ONE’s publication criteria as it currently stands. Therefore, we invite you to submit a revised version of the manuscript that addresses the points raised during the review process.

We look forward to receiving your revised manuscript.

Kind regards,

Pervez Alam

Academic Editor

PLOS ONE

Reviewers' comments:

Reviewer's Responses to Questions

**Comments to the Author**

1. Is the manuscript technically sound, and do the data support the conclusions?

Reviewer #1: Yes

Reviewer #2: Partly

2. Has the statistical analysis been performed appropriately and rigorously?

Reviewer #1: Yes

Reviewer #2: I Don't Know

3. Have the authors made all data underlying the findings in their manuscript fully available?

Reviewer #1: Yes

Reviewer #2: Yes

4. Is the manuscript presented in an intelligible fashion and written in standard English?

Reviewer #1: Yes

Reviewer #2: No

Reviewer #1: Although the manuscript has novelty, still some points need to be addressed before final publication:

1. There is a need to check citation again. some references are missing.

2. In introduction last paragraph not mentioned about section I, add details of section I also.

3. A robust assessment system is demonstrated by the use of the AHP approach to integrate RT, STI, and other acoustic data. Nevertheless, it is unclear from the discussion which other auditory factors were taken into account. Clarity and comprehensiveness would be enhanced by identifying and characterizing these characteristics explicitly.

4. Including sound absorption coefficients of wood fiberboard at different frequencies would strengthen the evidence for its effectiveness.

5. A deeper comparison of this study's methodology and results with those of Siregar I's team and Zhang C et al., highlighting differences in precision or applicability, would enhance the discussion.

6.The use of both simulation and actual measurements to validate results is commendable. It would be beneficial to elaborate on the simulation process, such as the tools/software used and how closely the simulations aligned with the actual measurements.

7.The discussion contains repetitive phrases such as "reverberation and reverberations of the sound," which could be streamlined for clarity. Additionally, some sentences could be restructured for better readability and logical flow.

Reviewer #2: In the manuscript, the author used the acoustic parameters such as reverberation time, language transmission index

etc. for a comprehensive score using AHP and optimized it through the ODEON software for a systematic evaluation

of the acoustics of a basketball stadium. The paper needs a serious rewriting both in terms of language and in terms

of explaining the results.

For example, in the related works section, after every reference the author wrote "The results showed that this

method has good practical", "... good effectivenss" etc.

"According to the linear system theory S(t)," does not mention what S(t)

". V The research basketball hall generally has a large volume" without mentioning V is the volume of the hall.

There are tons of places where the language needs correction.

The manuscript needs major improvements before publication. See specific comments below.

1. Fig. 4a, The author does not explain the spread of RT for 250 Hz.

2. Fig. 4a, I would expect the RT or any other observed quantity to be identical for the points that are symmetrical.

e.g. point 3 and 6, 1 and 7 etc. Also, why the author do not show any result for the other measuring points

(may be in the supplement)

3. The comprehensive score M in equation 15 is somehow used to optimize the acoustics, but the author failed

to mention it anywhere. Also, there is no figure associated with it to show the difference in before and after

improvement.

4. The figures (4a, 8 and ) do not require 3d axis and actually makes it difficult to read.

5. Figure 6a: What kind of material is used to generate the figure and how does it change for different materials?

6. The first two references are not mentioned in the manuscript

7. Why the source of sound is chosen to be in the center and how does it change if I have multiple sources at

different location with varying intensities? It will be easy to simulate different conditions and show that the

result is robust.

8. I do not understand the statement "The wall sound absorption material can protect the wall and column from

being damaged by the basketball impact, and provides better sound absorption effect." The ball hardly goes

outside the playing arena

9. Typo in first page "ODION" should be "ODEON", Pg. 8 the word "museum" should be stadium, I believe.

There are numerous other places that need to be fixed.

**Do you want your identity to be public for this peer review?** For information about this choice, including consent withdrawal, please see our Privacy Policy

Reviewer #1: No

Reviewer #2: No

---

## [Author Response · Author response to Decision Letter 1]

9 Mar 2025

Reviewer #1: Although the manuscript has novelty, still some points need to be addressed before final publication:

1. There is a need to check citation again. some references are missing.

Reply: Thank you for pointing out the existing issues. The reference citation status of the manuscript has been checked. The references [1] and [2] cited in the introduction have been added as follows:

However, due to the parallel arrangement of the walls in the basketball stadium, the echo and reverberation phenomena generated have caused serious interference to the sound environment, posing difficulties for teaching and competition [1].

Therefore, in-depth research on the impact of the sound environment of basketball stadiums on user experience and proposing corresponding acoustic design strategies have positive significance for improving the sound environment quality of basketball stadiums and enhancing user experience [2].

2. In introduction last paragraph not mentioned about section I, add details of section I also.

Reply: Thank you for providing the modification suggestions. The content of section I has been added to the introduction as follows:

Section I introduces the importance of the sound environment in basketball stadium and methods to improve its quality, and points out the specific methods of this study.

3. A robust assessment system is demonstrated by the use of the AHP approach to integrate RT, STI, and other acoustic data. Nevertheless, it is unclear from the discussion which other auditory factors were taken into account. Clarity and comprehensiveness would be enhanced by identifying and characterizing these characteristics explicitly.

Reply: Thank you for your valuable suggestion. In the discussion section, the scoring of sound pressure level in this research design has been supplemented, as follows:

In large spaces such as basketball stadiums, measuring sound pressure levels helps evaluate the intensity of sound. Before improvement, the scores for , , and were 1, 1, and 3, respectively. At this time, the comprehensive score is =1.21, which is within the range of 1.00 ≤ ≤ 1.50, and the sound environment evaluation of the basketball stadium is poor. After improvement, the scores of , , and have been increased to 3, 2, and 3, respectively. At this time, the comprehensive score is =2.74, which is within the range of 2.50 ≤ ≤ 3.00. The sound environment evaluation of the basketball stadium is good. Through improvement measures, the sound environment quality of the basketball stadium has been significantly improved.

4. Including sound absorption coefficients of wood fiberboard at different frequencies would strengthen the evidence for its effectiveness.

Reply: Thank you for your evaluation. The sound absorption coefficient of wood fiberboard at different frequencies has been supplemented to the bottom of Figure 6. Meanwhile, the discussion section has been supplemented with the impact of wood fiberboard on the acoustic environment performance inside the college basketball stadium. The content is as follows:

In Figure 6 (b), the sound absorption coefficient of wood fiberboard in the range of 125Hz~250Hz is between 0.2~0.3. Its sound absorption coefficient in the range of 500Hz~1000Hz is between 0.4~0.7. Its absorption coefficient is highest in the range of 2000Hz~4000Hz, between 0.7~0.9. The sound absorption coefficient of wood fiberboard at different frequencies directly affects the RT of the basketball stadium. By absorbing mid to high frequency sound waves, the wood fiberboard significantly reduces RT, keeping the RT value within a reasonable range between 500Hz-4000Hz. The problem of low low-frequency sound absorption coefficient is compensated for by combining perforated steel plates and ultrafine glass wool, achieving full frequency RT optimization.

The efficient absorption of mid to high-frequency sound waves by wood fiberboard significantly improved the STI, from 0.30 to 0.58. This further proves that the clarity of speech in the basketball stadium has been significantly improved.

5. A deeper comparison of this study's methodology and results with those of Siregar I's team and Zhang C et al., highlighting differences in precision or applicability, would enhance the discussion.

Reply: Thank you for providing the modification ideas. The comparison between this study's methodology and results and other methods and results has been supplemented, as follows:

Compared with other studies, Zhang C et al. proposed that fiber materials can achieve good sound absorption performance for reducing noise pollution [22]. The wood fiber board selected as the sound-absorbing material in this study not only has good sound absorption performance, but also plays an important role in the acoustic improvement of the basketball stadium, effectively reducing noise and reverberation, and improving the quality of the sound environment. In addition, the research method of Siregar I team is to use software and trial and error analysis to evaluate university auditoriums, and to use different acoustic materials for sound insulation design. The results indicate that this method can comprehensively adjust the indoor acoustic environment and ensure the functionality of indoor acoustics [23]. Similarly, this study conducted scientific simulations using ODEON software to carry out professional construction, ensuring the effectiveness and reliability of the sound environment renovation in the basketball stadium. From the perspective of improving the quality of the acoustic environment and the durability of facilities, the wood fiberboard and Perforated steel plates used in this study have high value and feasibility.

6.The use of both simulation and actual measurements to validate results is commendable. It would be beneficial to elaborate on the simulation process, such as the tools/software used and how closely the simulations aligned with the actual measurements.

Reply: Thank you for your commend. The content of the simulation process has been supplemented as follows:

The study used the 3D modeling software Sketchup to build a basketball stadium model, ensuring that the size, layout, and shape of the internal space accurately reflect the actual venue. The distribution of measuring points (measure point, MP) in the basketball stadium is shown as shown in Figure 2.

Simulate the propagation path of sound rays in the model in ODEON software, and release a set of acoustic particles in the simulation and track their propagation path.

7.The discussion contains repetitive phrases such as "reverberation and reverberations of the sound," which could be streamlined for clarity. Additionally, some sentences could be restructured for better readability and logical flow.

Reply: Thank you for your suggestion. The repeated phrases in the discussion have been simplified and the following sentences have been restructured. The modified statement is:

This study used the AHP combined with RT, STI, and other acoustic parameters to establish a comprehensive evaluation system, which effectively improved the acoustic environment of the basketball stadium. The results indicate that the improved RT values are below the international standard upper limit in all frequency bands, especially at 250Hz where the improvement effect is most significant, with RT reduced to 1.59s. This indicates that the RT of the basketball stadium has been significantly controlled, reducing reverberation and making the sound clearer and more distinguishable. The efficient absorption of mid to high-frequency sound waves by wood fiberboard significantly improved the STI, from 0.30 to 0.58. This further proves that the clarity of speech in the basketball stadium has been significantly improved. Yang D and other professionals used non-linear regression models to evaluate STI and RT values in classroom interiors, obtaining relatively accurate data [21]. In contrast, this study not only measured the acoustic environment of the basketball stadium, but also verified the results of STI through simulation, and achieved comprehensive improvement of the acoustic environment through optimization of multi band RT. The comprehensive score of 2.74 indicates the inadequacy of the basketball stadium's sound environment before the improvement, while the evaluation result after the improvement is favorable, showing that the measures taken have effectively enhanced the acoustic quality.

Reviewer #2: In the manuscript, the author used the acoustic parameters such as reverberation time, language transmission index etc. for a comprehensive score using AHP and optimized it through the ODEON software for a systematic evaluation of the acoustics of a basketball stadium.

The paper needs a serious rewriting both in terms of language and in terms of explaining the results.

For example, in the related works section, after every reference the author wrote "The results showed that this method has good practical", "... good effectivenss" etc.

Reply: Thank you for pointing out the existing issues. In the related works section, the problematic statements have been rewritten. The content has been modified as follows:

Shi W et al., for the performance evaluation of badminton stadiums, built a computer program combining hierarchical analysis through MATLAB to determine the key performance indicators of the venues and calculate the importance weight of the evaluated performance attributes. The improvement strategy for badminton stadium facilities was ultimately determined [4]. Lee H M et al., in order to study the acoustic environment of the university campus, they used questionnaire survey and real-time noise level measurement to classify sound sources, evaluate the surrounding environment of the campus, and compared the noise levels of various sound sources. The university with the best acoustic environment quality was evaluated as a result [5]. Zhang D et al. established an evaluation system for building ventilation, sound, lighting, and other related indicators to assess the indoor environmental quality of sports facilities. The results indicate that ventilation indicators and noise treatment are key factors in ensuring indoor quality [6]. Jung C et al., in order to evaluate the indoor environment of residential buildings, researchers completed the comprehensive evaluation according to the post-occupancy assessment survey and the weight of indoor environmental factors, and determined the priority of residents' space needs. It was found that thermal comfort factors in summer and indoor air quality factors in winter are more important. Di Loreto S et al. developed a tool for predicting classroom speech transmission metrics in order to evaluate the comprehensibility of speech in the classroom. It was found that this method can be applied to indoor sound environment design and has certain computational robustness [8]. Zhang L et al. designed a system using acoustic equipment and video sensors to evaluate indoor gymnastics activities and improve activity quality. The results indicate that the system can alleviate the data imbalance between acoustic and perceptual environments [9].

As an acoustic simulation tool, ODEON software has been widely used to improve the acoustic environment in various venues and show its potential in the gymnasium field. Murgia S et al. used ODEON software to analyze the indoor environment of elementary school classrooms and conducted comprehensibility tests. It was found that long RT and high background noise can have a negative impact on speech transmission [10]. Esmaeil and Salehi, in order to study the variables affecting the receiving sound intensity, professionals used the full factor method to identify the target scene of the variables, and used ODEON software to determine the frequency range with the greatest influence on the sound intensity. It was found that the frequency of sound and the distance between the sound source were inversely proportional to the sound intensity, and the octave band of sound had the greatest impact on the sound intensity [11]. Labia L et al., in order to improve the acoustic quality of the conference room, the team used ODEON software to complete the acoustic measurement of the conference room, simulate sound absorption and diffusion processing. Afterwards, determine the optimal number and location of acoustic panels indoors, and use absorbent materials appropriately to reduce costs [12]. Lv C et al. developed a spatial acoustic simulation system that combines Unity 3D and ODEON software to achieve model visualization. The results show that the system has good visualization effect and operability, and the absolute error of acoustic parameter values is small [13]. Sanz Soriano et al., for stage acoustics and parametric design, they enhanced the design collaboration in the early design stages through ODEON software and 3D modeling platform, and evaluated the impact of architectural changes on the reflection of supporting the orchestral ensemble. The different division of labor in these software can achieve real-time feedback and better integrate architectural and acoustic design [14]. Iannace G et al., in order to carry out the indoor sound school, they introduced the sound school system made of sound-absorbing flexible board to measure the acoustic characteristics of the empty room, and used ODEON software to analyze the acoustic characteristics and analyze the room configuration. This test found that installing 1.5mm soft wooden boards 3.0cm away from the room wall is reasonable [15].

"According to the linear system theory S(t)," does not mention what S(t) ". V The research basketball hall generally has a large volume" without mentioning V is the volume of the hall.

Reply: Thank you for your question. There are some translation errors here, the corrected sentence should be:

According to linear system theory, if an acoustic system in a closed space is linear, the output signal at the receiving point can be calculated from the sound source signal [17].

Basketball stadiums generally have a large volume , and at this time, the air has a greater absorption capacity for sounds with frequencies above 2000Hz.

There are tons of places where the language needs correction.

Reply: Thank you for your suggestion. The manuscript has been checked and the language has been corrected.

The manuscript needs major improvements before publication. See specific comments below.

1. Fig. 4a, The author does not explain the spread of RT for 250 Hz.

Reply: Thank you for your feedback. The reason for the spread of RT for 250 Hz has been explained as follows:

This is because low-frequency sound waves are easily affected by various factors such as reflection, standing waves, and sound wave diffraction when propagating in large spaces. The selected MP1, MP2, MP7, and MP8 are located in the corners of the rectangular field, and the positions of the other MPs also exacerbate this unevenness, resulting in significant differences in RT measured at different locations.

2. Fig. 4a, I would expect the RT or any other observed quantity to be identical for the points that are symmetrical.

e.g. point 3 and 6, 1 and 7 etc. Also, why the author do not show any result for the other measuring points (may be in the supplement)

Reply: Thank you for pointing out the existing issues. Your expected viewpoint is correct. The measurement points in Figure 4 (a) are actually MP5 and MP32� It's not an MP6 and MP31. The 8 measurement points selected for the study are the four corner points of a representative rectangular field (MP1, MP2, MP7, and MP8), as well as measurement points that are closer to the sound source (MP3, MP5) and farther away (MP10, MP32). Due to space limitations, only representative measurement points will be explained. The modified Figure 4 is as follows:

Figure 4. Comparison of RT and D50Results

In Figure 4 (a), the RT results of MP at 125Hz are less consistent and more scattered. This is because low-frequency sound waves are easily affected by various factors such as reflection, standing waves, and sound wave diffraction when propagating in large spaces. The selected MP1, MP2, MP7, and MP8 are located in the corners of the rectangula

---

## [Decision Letter · Decision Letter 1]

3 Sep 2025

Optimization Strategy for Sound Environment of College Basketball Stadium

PONE-D-24-35144R1

Dear Dr. Liang,

We’re pleased to inform you that your manuscript has been judged scientifically suitable for publication and will be formally accepted for publication once it meets all outstanding technical requirements.

Kind regards,

Mario Milazzo

Academic Editor

PLOS ONE

Additional Editor Comments (optional):

Based on the satisfaction of the Reviewer, the manuscript is provisionally accepted.

Reviewers' comments:

Reviewer's Responses to Questions

**Comments to the Author**

Reviewer #1: All comments have been addressed

Reviewer #2: All comments have been addressed

2. Is the manuscript technically sound, and do the data support the conclusions?

Reviewer #1: Yes

Reviewer #2: Yes

3. Has the statistical analysis been performed appropriately and rigorously?

Reviewer #1: Yes

Reviewer #2: Yes

4. Have the authors made all data underlying the findings in their manuscript fully available?

Reviewer #1: Yes

Reviewer #2: Yes

5. Is the manuscript presented in an intelligible fashion and written in standard English?

Reviewer #1: Yes

Reviewer #2: No

Reviewer #1: Very well addressed the comments. Appreciated for their hard work. The manuscript is accepted for publication.

Reviewer #2: (No Response)

**Do you want your identity to be public for this peer review?** For information about this choice, including consent withdrawal, please see our Privacy Policy

Reviewer #1: No

Reviewer #2: No

---

## [Editor Report · Acceptance letter]

PONE-D-24-35144R1

PLOS ONE

Dear Dr. Liang,

I'm pleased to inform you that your manuscript has been deemed suitable for publication in PLOS ONE. Congratulations! Your manuscript is now being handed over to our production team.

Kind regards,

on behalf of

Dr. Mario Milazzo

Academic Editor

PLOS ONE